# *MOSR* and *NDH*A Genes Comprising G-Quadruplex as Promising Therapeutic Targets against *Mycobacterium tuberculosis*: Molecular Recognition by Mitoxantrone Suppresses Replication and Gene Regulation

**DOI:** 10.3390/genes14050978

**Published:** 2023-04-26

**Authors:** Arpita Dey, Kushi Anand, Amit Singh, Ramasare Prasad, Ritu Barthwal

**Affiliations:** 1Department of Biosciences and Bioengineering, Indian Institute of Technology Roorkee, Roorkee 247667, Uttarakhand, India; 2Centre for Infectious Disease Research, Indian Institute of Science, Bengaluru 560012, Karnataka, India

**Keywords:** G-quadruplex, *mosR*/*ndhA* genes, downregulation, DNA replication, DNA targeting by mitoxantrone, spectroscopy and calorimetry, DNA stabilization, potent G-quadruplex ligand, *Mtb*

## Abstract

Occurrence of non-canonical G-quadruplex (G4) DNA structures in the genome have been recognized as key factors in gene regulation and several other cellular processes. The *mosR* and *ndhA genes* involved in pathways of oxidation sensing regulation and ATP generation, respectively, make *Mycobacterium tuberculosis* (*Mtb*) bacteria responsible for oxidative stress inside host macrophage cells. Circular Dichroism spectra demonstrate stable hybrid G4 DNA conformations of *mosR*/*ndhA* DNA sequences. Real-time binding of mitoxantrone to G4 DNA with an affinity constant ~10^5^–10^7^ M^−1^, leads to hypochromism with a red shift of ~18 nm, followed by hyperchromism in the absorption spectra. The corresponding fluorescence is quenched with a red shift ~15 nm followed by an increase in intensity. A change in conformation of the G4 DNA accompanies the formation of multiple stoichiometric complexes with a dual binding mode. The external binding of mitoxantrone with a partial stacking with G-quartets and/or groove binding induces significant thermal stabilization, ~20–29 °C in *ndhA/mosR* G4 DNA. The interaction leads to a two/four-fold downregulation of transcriptomes of *mosR*/*ndhA* genes apart from the suppression of DNA replication by *Taq* polymerase enzyme, establishing the role of mitoxantrone in targeting G4 DNA, as an alternate strategy for effective anti-tuberculosis action in view of deadly multi-drug resistant tuberculosis disease causing bacterial strains t that arise from existing therapeutic treatments.

## 1. Introduction

Tuberculosis (TB) disease is caused by pathogenic bacteria named *Mtb*, a family member of Mycobacteriaceae, and was the leading cause of approximately 1.3 million deaths in the year 2021. An extensive drug-resistant (XDR) and multidrug-resistant strain (MDR) makes it more vulnerable to society [1]. It is essential to have a thorough understanding of the mechanism of the bacteria’s survival and persistence pathways to uncover new therapeutic targets. A novel drug target, such as G-quadruplex DNA, may turn out to be a cornerstone for eliminating drug-susceptibility and multidrug-resistant tuberculosis. A G-quadruplex (G4) is a repetitive guanine-rich sequence that forms a secondary structure with a distinct pattern of motifs/loops and is an evolutionary conserved pharmacological target found in prokaryotes, eukaryotes, and viruses. In humans, the G4 DNA/RNA is associated with many vital cellular processes, such as replication, transcription, etc. In addition, this G-quartet is also found in the telomere, which inhibits telomerase enzyme or the promoters of several proto-oncogenes, such as *BCL-2, c-MYC*, *KRAS*, and *c-KIT*, making it a potential therapeutic target for cancer [2,3,4]. The RNA/DNA G-quadruplex structure is found in several bacterial proteins and enzymes involved in pathogenesis and antigen diversification. In *Mtb*, having a 65% GC-rich genome, bioinformatic analysis has identified more than a thousand motifs that have the potential to fold into G4 DNA structures [5,6,7]. 

Among the genes of *Mtb* with a predicted G4 in their promoter region, we selected two candidates for the experimental validations, namely oxidation-sensing Regulator transcription Factor (*mosR*) and membrane NADH dehydrogenase (*ndhA*) [5]. Many dormancy regulon factors control most of the mycobacterial hypoxic growth-related operons. *Mtb* shows an anaerobic response after forming the Δ*mosR* [8]. Under certain stress conditions in the host’s microenvironment (e.g., a high O^−^ level), bacteria go into the log phase and *mosR* expression becomes lowered as a resultant relief from the repression of other genes, such as hypoxia and phagosome expression genes. The *mosR* gene is, thus, responsible for survival in a stress condition and is involved in oxidative phosphorylation [8]. NADH dehydrogenase is a key factor enzyme for processing ATP synthesis, catalyzed by the F_1_F_o_–ATP synthase complex [9], which is an essential pathway in *Mtb,* regardless of the growth environment. Although both *ndh* and *ndhA* are genes with a biochemically similar property and both have a significant role in ATP synthesis by ATP synthase in the oxidative phosphorylation pathway, *ndh* does not have a putative G-quadruplex sequence. Importantly, while common among bacteria and protozoan parasites, these enzymes are absent from the mammalian genome, which further highlights their potential as drug targets [7].

Intramolecular G4 forming sequences, consisting of stretches of guanines (G-runs or G-tracts) separated by short gaps (loop sequences), can be predicted to fold into stable G4s under near-physiological conditions using Bio-informatics tools. The presence of three Gs in each island, and loops no longer than 11 nt that are expected to form stable G4s [5,6,10,11], G-score values above 19 indicate that the corresponding PQS (Predicted quadruplex sequences) is predicted to fold into a highly stable G-quadruplex structure [10,11]. The transcription start site (TSS) of *mosR* is at −38 Nts. The genome sequence has 3, 5, 3, 3 Guanines (“Gs”) in the four islands and loops of 7–9 nucleotides (nt). The corresponding transcription site (TSS) of *ndhA* is at −47 Nts. It has three Gs in each of the four islands and six bases in each loop (Appendix A). These features in *mosR*/*ndhA* sequences suggest the formation of stable G4s [5]. A QGRS mapper [10] provides a G-score = 72 and 67 for the *mosR* and *ndhA* genes, respectively. According to the PQS finder [11], the G-score for *ndhA* G4 is 52 and that for *mosR* G4DNA is 50. Although the G-score values are different due to the different algorithms used in the software, both of the G-score values are well above the threshold and, therefore, indicate the formation of stable G4s. Thus, both of the genes have a high putative G-quadruplex constructing sequence score, making them a favorable and efficient therapeutic target against the H37Rv deadly strain. The H37Rv strain was obtained from a patient’s sample and is a more susceptible strain. Since the isolation of the H37Rv strain in 1905, this strain of *Mtb* has found widespread use in biomedical research. Unlike some clinical isolates, it retains full virulence in animal models of tuberculosis; it is also drug resistant/susceptible and amenable to genetic manipulation [5,7,12,13,14].

A unique electron-rich aromatic surface made up of four guanine nucleotides is provided by a planar arrangement of the G-quartet structure, held together by the Hoogstein hydrogen bonding scheme. In addition, the groove binding pockets and loop arrangement are ideal for a specific ligand-based therapeutic design. The mode of action of the G-quadruplex depends on stability with the ligands and the basis of forming different structures (parallel, antiparallel and hybrid) in the presence of ions such as K^+^, Na^+^, Mg^+^, and the pH [15,16,17,18,19]. Several G-quadruplex ligands (e.g., anthraquinone derivatives, flavonoids, alkaloids, telomastatins, anthracyclines, etc.), bind through external groove binding mode [2,3,4,20,21]. Few others, such as RHPS4, daunomycin (an anthracycline) and epiberberine alkaloid bind through stacking mode at the ends of DNA or intercalate between the G-quartets via favorable π-π interactions between the aromatic surface of the G-quartets and ligands [2,3,4,17]. Aromatic end stackers and groove binding ligands increase the binding affinity by interacting with side chains. To the best of our knowledge, there are three reports of ligands tested against some of the genes comprising G4s in *Mtb* for their selectivity for towards tetraplex structures over duplex [5,6,7]. The ligands, BRACO-19 and c-exNDI 2, were found to stabilize *mosR*, *ndhA*, *zwf1*, and *clpx* gene sequences, inhibit *Mtb* growth and downregulate their gene expression [5]. The highly conserved potential of the G-quadruplex motifs (PGQs) in three essential genes, namely *espK*, *espB*, and *cyp5*, across 160 strains of the *Mtb* genome, suggested their role in bacterial survival and virulence [6]. A well-known G4 ligand, TMPyP4, was found to bind to and stabilize these PGQ motifs. TMPyP4 had an inhibitory role in the expression of *espK* and *espB* virulence genes and in the survival of *Mtb* [6]. The synthesized compounds CBR-1825, having a thioquinazoline (TQZ) core, and CBR-4032, having a tetrahydroindazole (THI) core, were found to inhibit ATP synthesis by targeting the *ndh-2* genes in *Mtb* [7].

We selected mitoxantrone (MTX) as a ligand for the present study on *Mtb* gene sequences. Mitoxantrone, 1,4-dihydroxy-5,8-bis[[2-[(2-hydroxyethyl) amino]ethyl] amino] anthracene-9,10-dione (Figure 1), known as a clinically approved anticancer drug, was found to target the human telomeric G4 DNA sequence leading to the inhibition of the telomerase enzyme [21]. MTX is a frequently used anthraquinone derivative in view of its non-cardiotoxicity because it lacks the amino sugar fragment and has a hydroxyquinone functional group with a polyamide side chain.

In the present study, cell-based assays and diverse biophysical tools have been used to predict and analyze the interaction of MTX with *mosR* and *ndhA* genes. A *Taq* polymerase and qRT-PCR assay show that MTX inhibits G-quadruplex formation and transcription via downregulation of the *mosR* and *ndhA* genes. The real-time binding of MTX to *mosR*/*ndhA* G4 DNA sequences was demonstrated using Surface Plasmon Resonance (SPR). An Absorption, Fluorescence and Circular Dichroism (CD) spectral analysis provided insight into the characteristics of ligand-binding and conformational changes. Finally, the ligand-induced thermal stabilization of the G4 DNA was determined from the melting profiles using CD spectroscopy and assessed independently with Differential Scanning Calorimetry (DSC). The present study corroborates a novel approach towards therapeutic investigations against *Mtb* infection by exploiting potential G-quadruplex targets.

## 2. Materials and Methods

### 2.1. Chemicals

Desalted oligonucleotide sequences, 34-mer d-TGGGCTAGCTCTAGGGGGCAGGGCTTTGACGGGT (*mosR*), 32-mer d-TGGGCCTTGTGGGCCTTGTGGGCCTTGTGGGT (*ndhA*) (Appendix A) and Mitoxantrone (MTX) were purchased from Sigma Chem Co., Burlington, MA, USA. The oligonucleotide sequences were dissolved in a buffer (pH 7.0) containing K^+^/Na^+^ and 1 mM EDTA. The stock solution of MTX was prepared by dissolving in KBEPS buffer (pH 7.0). The concentration of DNA sequence and MTX were determined using the molar extinction co-efficient ε = 69,800 M^−1^ cm^−1^ at 256 nm (per strand) and ε = 20,900 M^−1^ cm^−1^ at 659 nm, respectively. Further details are available in the Appendix A.

### 2.2. Surface Plasmon Resonance (SPR)

SPR experiments were conducted using the Biacore T200 optical biosensor system (GE Healthcare, Chicago, IL, USA). *mosR/ndhA* G4 DNA were immobilized onto a streptavidin-derivatized sensor chip, BIACORE SA (GE Healthcare Life Sciences, Little Chalfont, Buckinghamshire, UK) and dissolved in HEPES buffer at pH 7.4. MTX prepared in HEPES buffer was passed over the respective immobilized DNA cells and subsequently dissociated from the complex after passing the regeneration buffer. The response unit was evaluated after subtracting the corresponding flow cell 2/4 (immobilized DNA) from the reference flow cell 1/3 (HEPES buffer) to obtain the response from the bound ligand. An analysis and fitting of the data of the Response Unit were performed using the Biacore T200 evaluation software to obtain the Dissociation constant (*K_D_*) [20]. 

### 2.3. Absorption Spectroscopy

Varying concentrations of *mosR*/*ndhA* G4 DNA were mixed with a fixed concentration of MTX (3 μM) to reach mole equivalent ratios (D/N) of MTX Drug (D) to Nucleic acid quadruplex (N) in the range of 0.31–10. Absorbance spectra of each sample were recorded using a Bio UV-visible spectrophotometer (CARY 100, Varian, Palo Alto, CA, USA) equipped with a thermostatic cell holder and quartz cuvette (path length 1 cm) in the wavelength range of 200–800 nm [20,21]. The intrinsic binding constant was calculated using the standard equations described in the Appendix A along with an analysis of the data.

### 2.4. Steady-State Fluorescence

Steady-state fluorescence experiments were accomplished using a Fluorolog-3 Spectro-fluorimeter LS55 (Horiba Jobin Yvon Spex^®^, Kyoto, Japan). The samples used in the absorption spectra were excited (λ_ex_ = 610 nm) and the emission spectra were recorded in the wavelength range 640–800 nm [20,21]. The fluorescence quenching constant *K_SV_*, binding constant *K_b_* and binding stoichiometry (*n*) of the complex (the number of ligands binding to the DNA) were determined using standard procedures. 

### 2.5. Method of Continuous Variation (Job Plot)

To determine the stoichiometry of a binding event, the total concentration of MTX and G-quadruplex DNA were kept constant (3 μM) but their relative mole fractions were varied. The emission intensity at λ*_em_* = 678 nm was measured after exciting the reaction mixture at λ_ex_ = 610 nm [20,21]. The difference in fluorescence intensity of free MTX (F_0_) and its complex (F) with G4 DNA (ΔF = F − F_0_) was plotted as a function of the mole fraction of the MTX yields binding stoichiometry. 

### 2.6. Time-Resolved Fluorescence

Time-resolved fluorescence measurements were performed using the Fluoro-Cube^®^-Fluorescence lifetime system, (make HORIBA Jobin YvonSpex^®^, Piscataway, NJ, USA) using a quartz cuvette (path length 10 mm) operating in a time-correlated single-photon counting (TCSPC) mode. The samples were excited using a fixed-wavelength Nano LED (λ_ex_ = 640 nm) with a pulse duration of <200 ps [20,21]. The data were fitted using a re-convolution method provided with the DAS 6.3 software producing the best chi-square fitting values.

### 2.7. Circular Dichroism

Circular dichroism (CD) spectra were recorded on an Applied Photophysics (Model Chirascan, Leatherhead, UK) spectropolarimeter using a 1 mm path length quartz cell, which was equipped with a programmable temperature-controlled cell holder. All CD spectra of the samples were recorded in the wavelength range 200–700 nm with a 1 nm slit width at a 1 nm interval [20,21]. The generated spectra were plotted after baseline correction and smoothening using the Savitzky–Golay algorithm provided by Chirascan software.

### 2.8. Thermal Profiling (T_m_) Using Circular Dichroism (CD)

Thermal melting profiles were performed [20] with a temperature range from 25–95 °C at the rate of 1 °C/min using a Jasco J-1500 CD Spectrometer (Jasco, Tokyo, Japan) equipped with a MCB-100 Mini circulation Bath, Peltier unit-controlled cell holder, and a xenon lamp. 

### 2.9. Thermal Profiling (T_m_) Using Differential Scanning Calorimetry (DSC)

Thermal transitions from ordered quadruplex (helix) DNA to a disordered (strand) state were monitored by determiningthe excess heat capacity as a function of temperature using a VP DSC Micro-calorimeter (Microcal Inc., Northampton, MA, USA). The sample was scanned from 25 °C to 120 °C at a scan speed of 60 °C/h at approximately 34 psi pressure [20]. Thermograms were analyzed using the inbuilt VP Viewer software with Origin 7.0. The 3-state model of curve fitting was used to fit the raw thermograms of the unbound G4 DNA and its complexes. 

### 2.10. Taq Polymerase Stop Assay

The template DNA strand, containing *mosR* and *ndhA* and all primers were procured from Sigma Aldrich (Appendix A). A primer annealing region was added to the 3′ end of the *mosR* and *ndhA* oligonucleotide sequences for the assay. Additional T-flanking bases were inserted at both the 5′ and 3′ ends to isolate the 3′ end of the primer from the first base of the G4 component. The PCR method was performed in a 10 µL reaction containing a concentration of 1 μg/μL template DNA (100 mM KCl), 10 mM primer, 2.5 mM dNTPs, 5 Units of *Taq* polymerase (Thermo Fisher Scientific, Waltham, MA, USA) and different concentrations of MTX (1.56–50 μM). The reaction mixture was prepared in nuclease-free water. The negative control was considered to be no template in the reaction mixture, producing no product. The PCR-based amplification was carried out using an initial denaturation at 95 °C for 5 min, followed by 30 cycles of denaturation at 95 °C for 30 s, annealing and extension at 64 °C for 30 s, and the final extension at 72 °C for 50 s using a thermal cycler (model T100, Bio-Rad, Hercules, CA, USA); after the end of the reaction, the PCR tubes were kept at 4 °C. The amplified PCR products were then separated by agarose gel electrophoresis by mixing with 6× loading dye, resolved on a 3% *w/w* agarose gel, and an image analysis was conducted using a Gel Doc EZ imager (Bio-Rad, Hercules, CA, USA) by staining with ethidium bromide. 

### 2.11. Alamar Blue Assay (MIC Calculation)

The metabolic activity of *Mtb* strains in the presence of various concentrations of MTX was monitored using the reduction/oxidation indicator Alamar blue redox dye. Briefly, 100 μL of Middlebrook 7H9 broth supplemented with 10% (*v/v*) ADS and 0.2% glycerol was dispensed in every well of a sterile 96-flat-bottom well plate (Thermo Scientific, Mumbai, India), and serial dilutions (2-fold) of MTX were prepared in the plate directly. Bacteria were cultured in a 7H9−ADS medium and grown until the exponential phase (OD_600_-0.6). A total of 100 μL of inoculum from each strain (approximately 1 × 10^5^ bacteria/well) was added in triplicate wells. A growth control (*Mtb* without drugs) and a sterile control (drugs in media without bacteria) were also included. The 7H9 medium was added to all perimeter walls to avoid evaporation. The plate was sealed, covered and incubated at 37 °C. After six days of incubation, Alamar blue (10% of total volume) was added to each well, and the plate was further incubated for 1 day. A change from non-fluorescent blue to fluorescent pink color indicated a reduction in Alamar blue and the fluorescence readings were recorded. Fluorescence was measured in a Spectramax M3 plate reader with an emission wavelength of 590 nm after exciting at wavelength 530 nm and converted as a growth inhibition percentage. The intensity of the pink color is directly proportional to the amount of bacterial growth. The Minimum Inhibitory concentration, MIC, was defined as the lowest drug concentration that prevented 90% of growth.

### 2.12. Gene Expression by qRT-PCR

The wild type *Mtb* (H37Rv) strain was grown in a 7H9 medium supplemented with 1× ADS (Albumin Dextrose Saline) to an OD_600_-0.4 and exposed to 5X MIC of MTX for 1 h in a rotating incubator (180 rpm) at 37 °C. After treatment, the total RNA was purified as described [22]. After the purification and DNase treatment, cDNA was synthesized from 500 ng of total RNA using a script cDNA synthesis kit as per the manufacturer’s instruction. Gene-specific primers for *Mtb mosR* (FP2_KAAD_*Rv1049*RT-5-′CGAATGCGCTTGCTACACC-3′ and RP2_KAAD_*Rv1049*RT-5-′CCTTCCGACAGCGAGATCAC-3′) and *ndhA* FP1_KAAD_*Rv0392*cRT-5-′AGACGGTCACGTCGAAATTG-3′ and RP1_KAAD_*Rv0392*cRT-5-′GCCGAAGTAGGACTGCTGTG-3′) were selected (Appendix A) for RT-PCR (Step one Applied Biosystem). iQ^TM^ SYBR Green Supermix was used for the gene expression analysis and the data were normalized to 16S rRNA expression. The experiment was conducted with a minimum of two biological replicates and the fold change was calculated concerning the untreated control. 

## 3. Results and Discussion

### 3.1. mosR/ndhA DNA Sequences Adopt Stable G4 DNA Conformation

The folding and topology of selected 34-mer *mosR* and 32-mer *ndhA* DNA sequences (Appendix A) were assessed using CD spectroscopy, which can distinguish between different types of parallel/antiparallel stranded G4 DNA having an *anti* or *syn* glycosidic bond angle besides a single-stranded/duplex DNA conformation [15,16,17,18]. The CD spectra of the B-forms of DNA are characterized by a positive long wavelength band at about 260–280 nm and a negative band around 245 nm [16]. Typically, guanines in a parallel G-quadruplex that have an *anti* glycosidic bond conformation throughout the sequence exhibit a positive peak at 264 nm and a negative peak at 240 nm. On the other hand, guanines in antiparallel G-quadruplex strands with an alternate *anti* and *syn* glycosidic bond conformation in each DNA strand show a characteristically strong positive peak at 295 and 245 nm and a negative peak at 265 nm. In several G4 DNA folded structures, a mixed 3+1 hybrid structure with three parallel and one antiparallel strand have been observed with the formation of a double chain reversal loop. The 3+1 hybrid G4 conformation provides a strong positive CD band at 290 nm due to the alternating *anti* and *syn* glycosidic conformation between the top and middle G-tetrads. In addition, a positive shoulder around 268 nm and a small negative band at 240 nm, which are characteristic of non-alternating *anti* glycosidic bond rotation between the middle and bottom G-tetrads, are observed [15,16,17,18]. The observed CD spectra of the *mosR* sequence showed the signature peaks of a mixed type of G4 DNA conformation in K^+^ with two positive peaks at 265 and 288 nm and a negative peak at 240 nm (Appendix A). The molar ellipticity values increased with the concentration of K^+^ (0–150 mM) significantly, further supporting the G4 formation [19]. The *ndhA* G4 sequence also showed (Appendix A) a mixed type of G4 DNA conformation but with a relatively higher population of antiparallel G4 DNA, with maxima at 287 nm, a small negative band at 240 nm, and a positive band in the form of a shoulder at 260 nm in the presence of K^+^. The molar ellipticity increased with K^+^ concentration as observed for the *mosR* sequence. Notably, a mixed type of G4 conformation was displayed by both *mosR* and *ndhA* sequences in water; that is, in the absence of K^+^, indicating a high propensity to fold and stability. Thus, the selected sequences of *Mtb* effectively fold into G4 DNA conformations. 

The thermal melting profiles of *mosR* and *ndhA* G4 DNA, obtained by monitoring CD as a function of temperature (Appendix A), yielded several melting temperatures (*T_m_*) due to the presence of distinct partially folded/unfolded intermediate species in the melting pathway of the ordered to disordered 3+1 mixed hybrid G4 DNA conformation [20], and are consistent with that reported in the literature [5]. Ligand binding to G-quadruplex DNA depends upon several parameters, such as the nature of the cations, their concentration, pH, viscosity, the base sequence in DNA that are involved in the folding of single stranded DNA to form G4 DNA, etc. Therefore, we determined *T_m_* as a function of K^+^ concentration in the range 0–150 mM (Appendix A). The results showed an increase in the thermal stability of *mosR*/*ndhA* G4 DNA with K^+^ concentration. Subsequently, we selected 100 mM as the concentration of K^+^ cations, in a 10 mM KBPES buffer (pH 7.0) solution, for further studies; since our focus was on the study of the interaction with stable solution structures. Having established the existence of *mosR*/*ndhA* as stable G4 DNA structures, the next step was to determine if MTX binds to them, and to ascertain the binding process accompanying the changes in conformation and stability of G4 DNA.

### 3.2. Real-Time Binding of Mitoxantrone to mosR/ndhA G4 DNA

Surface Plasmon Resonance (SPR) is a reliable analysis of the kinetics parameters as well as the steady state of the ligand–macromolecule interaction [20]. Upon binding of the increasing concentration of MTX to *mosR* and *ndhA* in the range 0.03–3.0 μM, the sensograms (Figure 2A,B) show a steady-state response on binding (Figure 2C,D). The steady-state analysis yields the affinity constant, *K_b_* = 3.16 × 10^5^ M^−1^, whereas the analysis of the kinetics of association and dissociation yield *K_b_* = 3.85 × 10^5^ M^−1^ for the MTX-*mosR* G4 DNA complex (Appendix A). For the MTX–*ndhA* assembly, the binding constant from the steady-state and kinetics data was found to be *K_b_* = 1.17 × 10^6^ M^−1^ and *K_b_* = 1.27 × 10^7^ M^−1^, respectively (Appendix A). Real-time binding showing an increase in steady-state response with concentration indicates a specific interaction between the ligand and the G4 DNA molecules.

### 3.3. Binding Characteristics of Mitoxantrone-G4 DNA Complex

The assessment of the binding of a variable ratio of ligand (D) and target G4 DNA (N) sequence was indicated by the absorption spectra. MTX shows four distinct wavelength maxima, λ_max_ = 242, 276, 609 and 660 nm [21]. A change in absorbance in the visible region was used for the analysis, as DNA does not contribute to the absorbance in this region. Upon the stepwise addition of varying concentrations of *mosR* G4 DNA to a fixed concentration of MTX (7 μM), the intensity of both absorption bands of MTX showed hypochromism of ~35% at D/N = 5.0 accompanied by a red shift, Δλ_max_ = 16 and 18 nm at 609 and 660 nm, respectively. Further addition of *mosR* resulted in hyperchromism with no further change in the wavelength maxima, the increase in absorbance being significant, up to ~75% at D/N = 0.5 at 678 nm (Figure 3A). In the case of the *ndhA* G4 DNA, similar hypochromism of ~29% at D/N = 5.0, accompanied by a red shift of 15 and 18 nm at 609 and 660 nm, respectively, followed by hyperchromism at D/N = 0.5, was observed (Figure 3B). This demonstrates the presence of two binding modes, one at D/N > 5.0 while the other exists at lower D/N ratios, in which most of the MTX is expected to be bound at preferential sites on the G4 DNA. The absence of an isobestic point indicates the existence of more than one stoichiometric complex. The binding characteristics are found to be similar to that observed with tetramolecular parallel stranded G4 DNA [21]. In the absence of a large red shift of ~45 nm, which is characteristic of classical intercalation [23], the predominant mode of binding to *mosR* and *ndhA* G4 DNA sequences is expected to be partial stacking via the insertion of chromophore of MTX in the vicinity of base quartets or end stacking with G-quartets in addition to external groove binding of the side chains [21]. A detailed analysis of the absorbance data (Appendix A) is indicative of the stoichiometry of the MTX–G4 DNA complex as 1, 2, 4 and 1, 2 in the MTX–*mosR* and MTX–*ndhA* complexes, respectively, which may arise from simultaneous stacking at the ends and groove binding near loops. An estimate of the intrinsic binding constant is *K_b_* = 1.6 × 10^4^ M^−1^ and 1.6 × 10^6^ M^−1^ at D/N = 0.36–2.00 and 2.00–3.80, respectively, for the *mosR* complexes, while the corresponding value for the *ndhA* complexes is *K_b_* = 1.0 × 10^6^ M^−1^ and 8.7 × 10^4^ M^−1^ at D/N = 0.36–3.80 and 3.80–8.75, respectively (Appendix A). The Scatchard plots were found to be nonlinear and could not be used to determine the binding affinity as they did not fit into a combination of 2–3 straight lines. The intrinsic binding constants may be considered as a rough estimate due to the existence of multiple binding sites, neighbor exclusion effects, and ligand–ligand interactions [21]. The existence of multiple binding sites resulting in the absence of a clear isobestic point has previously been reported in the literature [20].

MTX exhibits well-resolved emission spectra with a maxima of λ_em_ = 695 nm for the excitation wavelength, λ_ex_ = 610 nm [21]. The addition of *mosR*/*ndhA* to a fixed concentration of MTX (7 μM) quenched the fluorescence by ~28% with a red shift of Δλ_em_ = 15 nm at D/N = 5.0. Subsequent stepwise additions of *mosR*/*ndhA*, however, resulted in a rapid increase in fluorescence by 160% and 90% in the *mosR* and *ndhA* complexes, respectively, at D/N = 0.3, with no further shift in the emission maxima (Figure 4A,B). Fluorescence quenching followed by an enhancement in intensity suggests the presence of multiple modes of binding, which is like the corresponding absorbance data. The iso-emissive point was not observed throughout the course of the titrations. An analysis of the fluorescence data [20,21] indicated the presence of several stoichiometric complexes with *n* ~1, 2, and 4 (Appendix A), which may arise from the binding of 1–4 molecules of MTX in different binding modes and positionings with respect to the G4 DNA. The *K_SV_* and *K_b_* obtained were close to that obtained from the absorbance data. The non-linear fit of the emission data at 695 nm showed the presence of two independent binding sites for the MTX–*mosR* and MTX–*ndhA* complexes, yielding two binding constants (Appendix A). The bimolecular quenching constant, *K_q_* ~2.6 × 10^13^–3.1 × 10^15^ M^−1^s^−1^, obtained using *K_sv_* = 1.3 × 10^4^–2.5 × 10^5^ M^−1^ and the fluorescence lifetime τ = 0.11–0.51 ns (discussed later) was much greater than the collision constant for a biological macromolecule and small molecule as a ligand, i.e., 2 × 10^10^ M^−1^s^−1^ [21], which rules out the possibility of any dynamic quenching. The presence of static quenching owing to ground state interactions was also evident from the accompanying changes observed in the absorbance and practically no change in the fluorescence lifetime. The initial quenching of the fluorescence at D/N = 5 may be attributed to the proximity of the MTX to the G4 DNA molecule, upon binding externally through partial stacking/end stacking/groove binding, which could bring about the efficient electron transfer from the nucleoside to the anthraquinone moiety. Qu et al. [24] have shown that the electron transfer process for daunomycin in the presence of d-GTP, an anthracycline having a conjugated aromatic chromophore, is found to be thermodynamically favorable, particularly for guanosines (∆*G* ~0.71 eV for excited singlet) and is a function of both proximity (~8–10 Å) and the relative orientation of the two molecules [21,24]. At low D/N ratios, MTX binds to the G4 DNA at a preferential site via stacking/partial intercalation/grooves in which it is apparently screened by a solvent, as demonstrated by Lin and Struve [25] in their studies of the fluorescence lifetime of MTX in aprotic solvents, resulting in the significant increase in fluorescent emission [21]. This is similar to the properties of several aromatic compounds (e.g., EthBr), for which the fluorescence is strongly quenched in an aqueous solution, but in a nonpolar or a rigid environment, a striking enhancement is observed.

Time-correlated single-photon counting (*TCSPC)* studies yielding the fluorescence lifetime may reflect upon the mode of binding of the ligand to DNA. The fluorescence decay of free MTX (3 μM) is bi-exponential, yielding the lifetime values, τ_1_ = 0.11 ns (90% population of major component) and τ_2_ = 0.47 ns (~10% population of minor component), which are close to that reported in the literature [21,25]. The fluorescence decay profiles of MTX changes on binding to G4 DNA indicating an interaction between the ligand MTX and the *mosR*/*ndhA* G4 DNA. The complex of MTX with *mosR* at varying D/N ratios showed an increase in the population of the minor component up to 50% while the individual lifetime values (τ_1_ = 0.09–0.21 ns and τ_2_ = 0.36–0.51 ns) remained practically the same (Appendix A). The decay profile of the MTX–*ndhA* complexes is like that of MTX–*mosR* complexes (Appendix A). The MTX chromophore may be shielded from the solvent after binding through partial stacking/end stacking/groove binding with *mosR/ndhA* G4 DNA leading to an increase in the higher lifetime component and preventing the fast mode of decay through water. This is consistent with the reported higher lifetimes prevalent in aprotic solvents [25] as well as the increase in intensity of the fluorescence at low D/N ratios [21]. 

The observed lifetimes of complexes suggest external binding/partial stacking of MTX to G4 DNA involving contributions from side chains, since the classical intercalation between G-quartets would lead to an increase in the lifetime, which is about three to four times that of the free ligand; e.g., the lifetime of ethidium bromide, a classical intercalator, in a free state (τ = 1.6 ns), shows an increase to τ = 22.2 ns upon binding to duplex DNA [21]. It has also been shown that the fluorescence lifetime of porphyrins, TMPyP4 and TrPyP4 (τ = 4.6–5.0 ns in the free form), increases to 6.6–7.5 and 10.6–12.4 ns, respectively, and corresponds to end stacking and groove binding modes of interaction with d-(TTGGGGT)_4_ G4 DNA [26]. 

Since the binding of MTX to G4 DNA showed the formation of more than one bound complex, we established the stoichiometry independently with the continuous variation method (Job plot) using fluorescence. An analysis of the Job plot (Figure 5A,B) indicates several inflection points between approximately linear regions [20,21].

Multiple complexes with a stoichiometry of 0.5:1, 1:1, 2:1 and 4:1 exist in solution, which corresponds to slope changes at the MTX/mole equivalent of G4 DNA fractions of 0.33, 0.50, 0.67, 0.82 (Figure 5A) and 0.33, 0.5, 0. 82 (Figure 5B) in *mosR* and *ndhA* complexes, respectively. This is consistent with the absorption data of *mosR* complexes, in which the plot of the reciprocal of absorbance (1/A) as a function of D/N at 678 nm showed inflection at D/N = 1.2, 2.0, and 3.7, suggesting a stoichiometry of 1:1, 2:1, and 4:1, respectively (Appendix A). The stoichiometry of 0.5:1, corresponding to a slope change at 0.33 (Figure 5A), is not obvious in the absorbance data, presumably due to fewer data points available in that region. In the same way, the results of the Job plot are found to be in accord with the absorption and fluorescent data of all other complexes.

### 3.4. Ligand-Induced Conformational Changes in mosR/ndhA G4 DNA

Upon the stepwise addition of MTX to the fixed concentration (20 μM) of *mosR* G4 DNA, the intensity of the CD band decreased continuously, by ~67% and 50% at 266 and 288 nm, respectively, up to D/N = 5.0 (Figure 6A). The negative CD band at 246 nm also decreased in magnitude significantly by ~73% up to D/N = 5.0. The proportion of antiparallel stranded G4 conformation increased or, in other words, equilibrium in the mixed hybrid conformation shifted towards an antiparallel conformation. Equivalent results have been observed in the case of binding to *ndhA*, in which the change in intensity of the bands was ~58% and 50%, at 265 and 288 nm, respectively, up to D/N = 5.0 (Figure 6B). The nonlinear fitting of CD as a function of ligand concentration gives an estimate of the affinity constants (Appendix A). No induced CD (ICD) band was observed in the wavelength range of 600–690 nm, unlike that observed in the binding of MTX to parallel stranded telomeric G4 DNA sequences, in which a bisignate [21] or negative ICD [27] were observed. The overall spectral features characteristic of G4 DNA remain unchanged on interaction with a ligand, which together with the absence of a negative induced CD band characteristic of classical intercalation, suggest the external binding of MTX to the grooves/loops or end stacking/partial stacking with guanine quartets of G4 DNA [28,29,30].

### 3.5. Ligand-Induced Thermal Stabilization of mosR/ndhA G4 DNA

The thermal denaturation profiles of ordered to disordered 100 μM *mosR*/*ndhA* G4 DNA sequences were monitored using Differential Scanning Calorimetry (DSC), which yields the melting temperatures (*T_m_*) by measuring the changes in heat capacity as a function of temperature. The DSC data was deconvoluted and found to fit into three “two state” transition models, yielding three melting temperatures (*T_m_*), indicating the existence of two intermediate species in the melting pathway. The three melting temperatures *T_m1_*, *T_m2,_* and *T_m3_* are centered at 47.3, 58.3, 71.4 °C and 46.8, 64.0, 84.4 °C for the *mosR* and *ndhA* G4 DNA sequences, respectively (Figure 7A,C). Upon the binding of MTX to *mosR* G4 DNA at D/N = 5.0, it is observed that *T_m_* increases, yielding thermal stabilization at ∆*T_m1_* = 19.9 °C, ∆*T_m2_* = 25.3 °C and ∆*T_m3_* = 28.8 °C, while the corresponding stabilization in *ndhA* G4 DNA is ∆*T_m2_* = 17.9 °C and ∆*T_m3_* = 10.1 °C (Figure 7B,D). These values of ∆*T_m_* (Table 1) are comparable to that obtained for other ligands to *mosR*/*ndhA* and other G4 DNA [5,6]. 

Structure-dependent melting profiles from the folded to unfolded conformation of *mosR*/*ndhA* G4 DNA in the native and complexed state at D/N = 5.0 were obtained from CD measurements at 265/260 and 287/288 nm with respect to the temperature at 2 °C intervals. The melting transitions of free *mosR* G4 DNA, yielded *T_m_* values at 25, 32, 54, 68, 85 °C and 28, 43, 60, 74, 87 °C at 288 and 265 nm (Figure 8A,C), respectively, which correspond well to the *T_m_* value 47.3, 58.3, 71.4 °C observed in the DSC thermograms as well as that reported in the literature [5]. Similarly, the *T_m_* values 28, 39, 45, 60, 84 °C and 26, 38, 60, 85 °C observed at 287 and 260 nm, respectively, for *ndhA* G4 DNA (Figure 8B,D), correspond to the 46.8, 64.0, 84.4 °C obtained from the DSC thermograms. Upon complex formation at D/N = 5.0, *T_m_* is found to increase in both complexes, yielding a thermal stabilization of ∆*T_m_* up to ~26 and 20 °C in *mosR* and *ndhA* complexes, respectively (Figure 8A–D; Appendix A). We observed that the melting pathways observed at two wavelengths, 265/260 and 288/287 nm, were different, giving rise to different ∆*T_m_* values. This is expected as melting transitions involve the reordering of a mixture of parallel/antiparallel hybrid conformations, which contributes differently to CD at 265/260 and 288/287 nm. The same has also been reported in the literature on the binding of ligands, BRACO-19 and c-exNDI 2 to *mosR*/*ndhA* G4 DNA [5] as well as 2,6 disubstituted anthraquinones to human telomeric G4 DNA [20]. It has been shown that the groove binding of mitoxantrone in monomeric/dimeric form to parallel d-[TTGGGG]4 [21] and d-[TTAGGG]4 [27] G4 DNA increases thermal stability by ~25 °C and, consequently, inhibits telomerase enzyme activity. The observed ∆*T_m_* obtained on the external groove/end stacking of mitoxantrone to *mosR*/*ndhA* G4 DNA are comparable to these results. 

### 3.6. Stall Replication Machinery by Taq Polymerase Enzyme Assay

To determine if MTX can induce G4 formation in the 34/32-mer *mosR*/*ndhA* DNA sequence (Appendix A) under physiologically relevant conditions, we performed a *Taq* DNA polymerase stop assay using a template containing *mosR*/*ndhA* sequences, to which 4 T were added at the 5′end and 3′end, followed by the addition of another 30 bases for annealing purposes at the 3′end, which are complementary to the selected *Taq* primer sequence of 30 bases (Appendix A). A G4 structure formed in the template strand can block the *Taq* polymerase synthesis of the complementary strand, since the enzyme requires single stranded DNA for its action. This G4-specific block can then be precisely solved in a denaturing polyacrylamide gel in terms of the intensity and sequence position. The technique is, thus, used to excise the ability of the ligand to stabilize the G4 structure in order to arrest the movement of the *Taq* polymerase enzyme at G-quadruplex sites. After the binding of MTX to the *mosR* and *ndhA* quadruplex DNA with the increasing concentration of MTX, the extension of the *Taq* polymerase enzyme became impeded. As a result, the shorter band was deferred to decreasing band intensity according to the concentration of the MTX binding to the template DNA (*mosR* and *ndhA*) (Appendix A); whereas it was also noted that neither a shorter product nor a small band was observed in the negative control (absence of DNA). It can be consummated that MTX strongly binds to the oligonucleotides’ template and inhibits further enzymatic processes.

### 3.7. Mitoxantrone Regulates Gene Expression of mosR and ndhA

In order to assess the involvement of G-quadruplex motifs in the regulation of gene expression upon treatment with MTX, the expression of *mosR* and *ndhA* genes in *Mtb* was checked using quantitative real-time polymerase chain reaction (qRT-PCR). Quantification of the gene transcripts was performed in relation to the 16S rRNA gene transcript (a housekeeping gene), and the Cq for non-template controls was >35 cycles. The results (Figure 9) show that compared to an untreated culture control, transcription of the *mosR* and *ndhA* genes was downregulated by a factor of 2–4-fold after the *Mtb* cells were treated with the 5XMIC value of MTX. The MIC of MTX was determined by treating the *Mtb* with various concentrations of MTX for six days. The MIC_90_ of MTX is 100 micromolar. To alleviate the general effect of MTX on gene expression, we performed qRT-PCR on the RNA derived from *Mtb* exposed to 5X MIC for a short duration (1 h). We confirmed the viability at different concentrations (in the range of 10X MIC–0.3X MIC) for 3 h and did not observe any cytotoxic effect of MTX until 10X of MIC (Appendix A). Further, we examined the expression of a few other genes that either lack (*Rv1403c*) or contain (*hupB*) the G4 motif in their regulatory regions. Like *ndhA* and *mosR*, we observed a significant downregulation of *hupB* (G4 motif containing genes), but not of *Rv1403c* (non-G4 containing genes) (Appendix A). The involvement of a G4-mediated mechanism in the transcription of *mosR* and *ndhA* suggest a therapeutic potential of the drug for *Mtb*, potentially through the inhibition of transcription by binding to the G4 motif.

### 3.8. Ligand Binding-Induced Thermal Stabilization Relates to Gene Function

The formation of the G4-motif inside cells is tightly linked to survival, stress resistance, and the modulation of gene expression in various organisms, e.g., humans, pathogenic bacteria, and viruses. The present investigations revealed that mitoxantrone has a specific affinity towards the G-quadruplex DNA motifs present in the regulatory region of *mosR* (−38 from TSS or −69 bp to −38 bp of the gene) and *ndhA* (−47 bp from TSS or −106 bp to −77 bp of the gene) under in vitro conditions. The external binding to G4 DNA induced significant thermal stabilization up to 29 and 20 °C in the *mosR* and *ndhA* G4 DNA sequences, respectively. A *Taq* polymerase stop assay demonstrated that the G4 structure assembly and replication machinery was hindered in the presence of MTX. Finally, a transcriptome analysis of the *mosR* and *ndhA* genes showed that gene expression was significantly reduced upon MTX treatment as compared to the untreated control, which may be attributed to the binding-induced thermal stabilization of *mosR*/*ndhA* G4 DNA, which is similar to ligand-induced thermal stability of G4 DNA inhibiting telomerase enzyme activity, requiring single stranded telomeric DNA for its function [2,3,4]. This implies a significant role of the transcriptional regulation of G-quadruplex in the regulatory region of these two genes. Moreover, these findings also support the idea of implementing the G-quadruplex ligand as a new therapeutic approach to treat this fatal human bacterial disease in combination with current anti-tuberculosis drugs. 

For drugs that target G4 DNA in cells, ligand-induced thermal stabilization is an important parameter since it would deplete the availability of single stranded DNA in telomeric DNA or the promoters of oncogenes due to the formation of stable G4 folded structures, which are required for their functioning and lead to apoptosis. Thus, the extent of the thermal stabilization of G4 DNA has emerged as an important therapeutic index besides the high affinity of binding. We observed that the binding affinity of mitoxantrone towards *ndhA* G4 DNA was higher than that for *mosR* G4 DNA. On the other hand, the thermal stability of the *mosR* complex was significantly higher than that of the *ndhA* complex. This is understandable, since the ligand–G4 DNA interaction involves short contacts between specific chemical groups/moieties of the ligand and G4 DNA, contributing to van der Waals, hydrophobic and hydrogen bonding energy terms, which define the stability of the complex. Unlike the well-characterized X-ray crystallographic and NMR [17,18] conformation of the 3+1 hybrid G-quadruplex of human telomeric DNA sequences (HTel22), the topology of *mosR* and *ndhA* G4 DNA has not been elucidated in the literature. Thus, the specific groups exposed and available for the interaction and accommodation of the approaching ligand are not spelled out. The aromatic chromophore of MTX may partially stack at the top/bottom/sides of G4 DNA. The three hydroxyl groups at the 1, 4, 14 positions, two oxygen atoms at the 9,10 positions and 11NH/12NH in the anthraquinone ring and side chains of MTX (Figure 1) offer immense possibilities for hydrogen bonding interactions with several groups in G4 DNA. Phosphates in the DNA backbone, G2NH_2_/O6/N7 of the G-quartets and the A/T base atoms available in the grooves/loops of G4 DNA have previously been reported to participate in ligand–G4 DNA interactions in the case of metalloporphyrin, aryl ethynyl anthraquinones, benzimidazole and furan derivatives [28,29,30,31]. For example, a detailed study of the binding of metalloporphyrins to different parallel/antiparallel G4 DNA structures (PDB ID:143D, 1KF1, 2HY9) by molecular dynamics simulations for 100 ns, using a FF12SB forcefield available in the AMBER12 package, revealed [29] how the saddle shape of porphyrin together with the d-TTA loop structure of antiparallel G4 covers the G-quartets at the top, bottom and grooves. Such possibilities can exist with MTX, with additional short polar contacts through its side chains, which will give rise to multiple binding sites, binding modes, and the stoichiometry (*n* = 1,2, and 4) observed in the present study. It may be noted that any aromatic chromophore can stack with a G-quartet with a non-specific interaction, but external contact at the grooves/loops is significant for ligand recognition and contributes to the interaction’s specificity. The exact nature of the interactions on external binding involving specific groups of ligands engaged in interactions, which are based on both the kind of ligand and the structural subtleties of the quadruplex, would dictate the stability of the ligand–G4 DNA complex. Differences in structural configuration contribute to thermal stability and more significant structural knowledge is required to comprehend the stabilization difference between the *mosR*/*ndhA* complexes and, hence, in evaluating the therapeutic potential of G4-binding ligands.

The anti-cancer action of mitoxantrone has been attributed to its binding to the DNA duplex by intercalating between the base pairs of DNA. Its binding affinity with calf thymus DNA in the presence of a Na^+^ [32] and Tris–HCl solution [33] is reported to be, *K_b_* = 6 × 10^6^ M^−1^ and 4 × 10^5^ M^−1^, respectively. Upon binding with a Herring testes DNA duplex (*K_b_* = 5.1–6.3 × 10^6^ M^−1^ at 298.15 K), mitoxantrone was found to increase the thermal stability of DNA by 11.75 K [34]. Our earlier studies with hexanucleotide d-(CTCGAG)_2_ yielded an equilibrium constant for binding at low (1.0–0.2) and high (28.0–1.1) mitoxantrone drug to DNA (D/N) ratios as *K_b_* = 1.8 × 10^5^ M^−1^ and 1.38 × 10^6^ M^−1^, respectively [35]. Further, binding with intermolecular parallel [d-(TTGGGGT)]_4_ G4 DNA with an intrinsic binding constant, *K_b_* = 0.6 × 10^6^ M^−1^, 3.5 × 10^6^ M^−1^, and 3.7 × 10^6^ M^−1^ for the range of D/N = 0.21–1.3, 1.16–4.75, and 4.75–9.5, respectively, yielded a thermal stabilization of Δ*T_m_* = 25 °C. The experiments using nuclear magnetic resonance spectroscopy clearly showed non-intercalative binding with no evidence of the opening of base quartets to accommodate the mitoxantrone chromophore [21,27]. In the present study, we observed groove binding to the intramolecular *mosR* hybrid G4 DNA sequence leading to a thermal stabilization up to 29 °C (Table 1). Thus, although the binding affinity of MTX with G4 DNA is comparable to that with a duplex DNA sequence [32], the binding mechanism is distinctly different. Its mode of binding to G4 DNA is non-intercalative external groove binding and the thermal stabilization is significantly larger than that of duplex DNA, which is an important therapeutic index.

Several questions about the selectivity, specificity and biological implications of G4 ligands and G4 aptamers remain unanswered, but the evidence already available reveals that G-quadruplexes are valuable and inventive pathogen-fighting tools. Excitingly, the fact that some G4 ligands have reached the clinical trial phase is encouraging. The G-quadruplex field of research would benefit from in vivo proof and structural data from both the physiologically relevant folding of the target G4 structures and their ligand interactions. In this context, our study emphasizes the understanding of an improved comprehension of the G4/i–motif/R-loop interaction that could lead to the development of novel ligands capable of modulating the expression of genes responsible for regulating the pathogenesis of *Mtb*. There is immense scope towards the discovery of novel anthraquinone-based compounds having slight modifications in their side chains, in the form of the nature of the substituent, and the substituent’s position on the anthraquinone ring [20], as reported in studies of the inhibition of telomerase enzymes to provide better efficacy. A question remains about the transcription factors associated with *mosR* and *ndhA* genes that may be disseminated because of MTX binding to the G quartet and stabilizing the structure. A comprehensive investigation of the high-resolution structure of this complex in the future may enlighten the necessary scaffolds and mechanisms. Based on the outcomes of the above study, we can, therefore, conclude that anthraquinones may be considered a novel target for the development of anti-tuberculosis therapy. 

## 4. Conclusions

We have shown that the virulence-determining *mosR*/*ndhA* genes of *Mtb* fold as a stable hybrid form of G-quadruplex DNA. Diverse biophysical techniques, that is, surface plasmon resonance, absorption, fluorescence and circular dichroism spectroscopy, demonstrate that mitoxantrone binds to the G4 DNA sequences with an affinity ~10^5^–10^6^ M^−1^, forming multiple stoichiometric complexes. Experimental evidence points towards the binding of MTX with G4, forming DNA sequences at the grooves/loops/ends externally, inducing a conformational change that is energetically favorable due to significant thermal stabilization (20–29 °C), as monitored by CD and heat capacity. The interaction leads to an inhibition of *Taq* polymerase enzyme action and the two/four-fold downregulation of *mosR*/*ndhA* gene expression. The inhibition of the expression of both oxidative stress-prone genes may be attributed to the binding and subsequent stabilization of their associated G4 DNA structures. The results on the binding affinity, thermal stabilization and the downregulation of expression are encouraging from a therapeutic point of view and show that mitoxantrone, or similar anthraquinone-based compounds, can be used as a promising strategy for developing highly selective G-quadruplex targeted therapies towards tuberculosis. Nevertheless, the present investigations pave the way for the discovery of novel anthraquinone-based compounds as potent G4 ligands.

## Figures and Tables

**Figure 1 genes-14-00978-f001:**
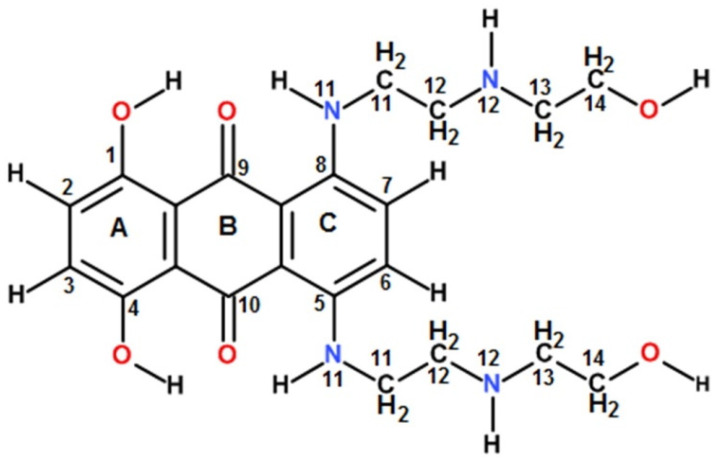
Schematic representation of Mitoxantrone (MTX).

**Figure 2 genes-14-00978-f002:**
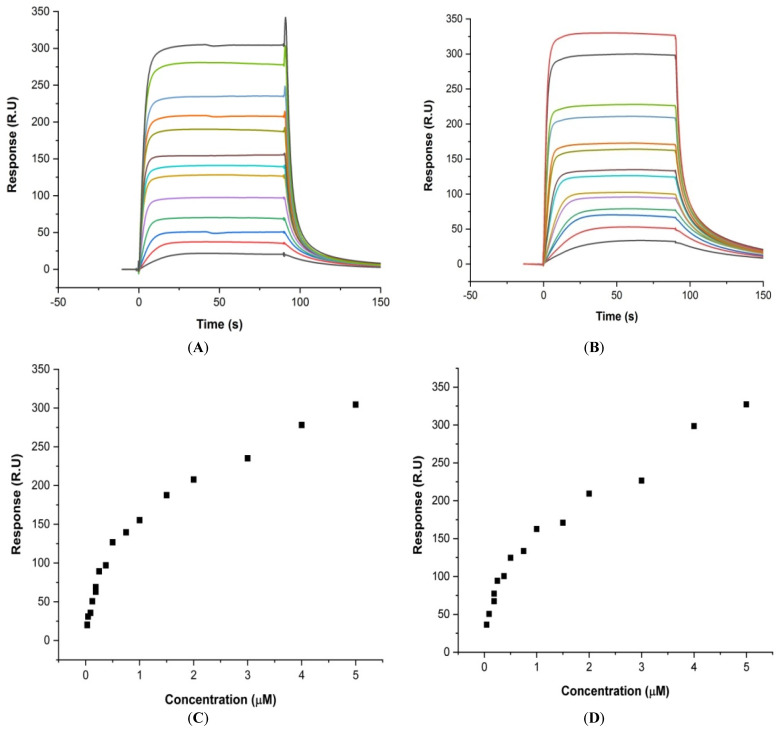
Results of the SPR for the interaction of MTX with G4 DNA using a HEPES buffer (pH 7.4) with 100 mM KCl at 25 °C. Sensograms obtained for the increasing concentration of MTX from 0.03 μM (bottom) to 3.0 μM (top): (**A**) *mosR* and (**B**) *ndhA*. Plot of steady-state binding showing the Response Unit (R.U.) as a function of the concentration of MTX: (**C**) *mosR* and (**D**) *ndhA*.

**Figure 3 genes-14-00978-f003:**
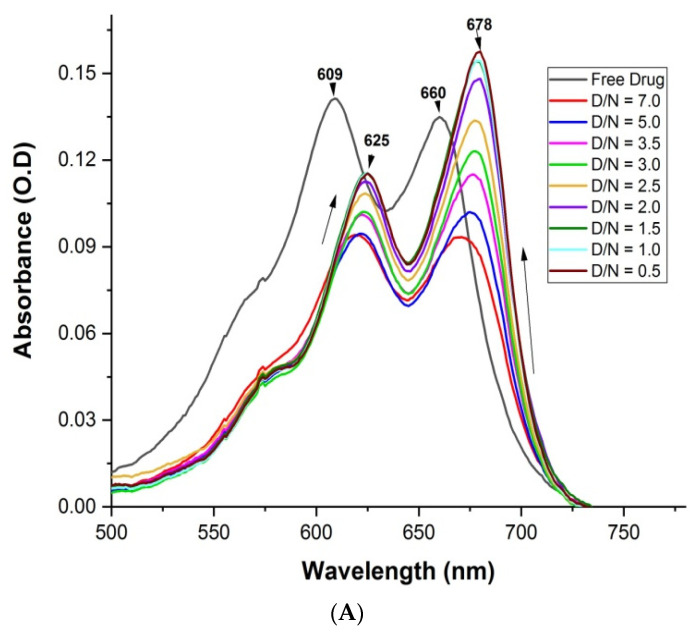
Absorption spectra of 7 μM MTX with increasing concentration of G4 DNA at D/N = 0.5–7.0 in 10 mM of KBPES buffer (pH 7.0) containing 100 mM KCl at 25 °C in the visible region: (**A**) *mosR* and (**B**) *ndhA*.

**Figure 4 genes-14-00978-f004:**
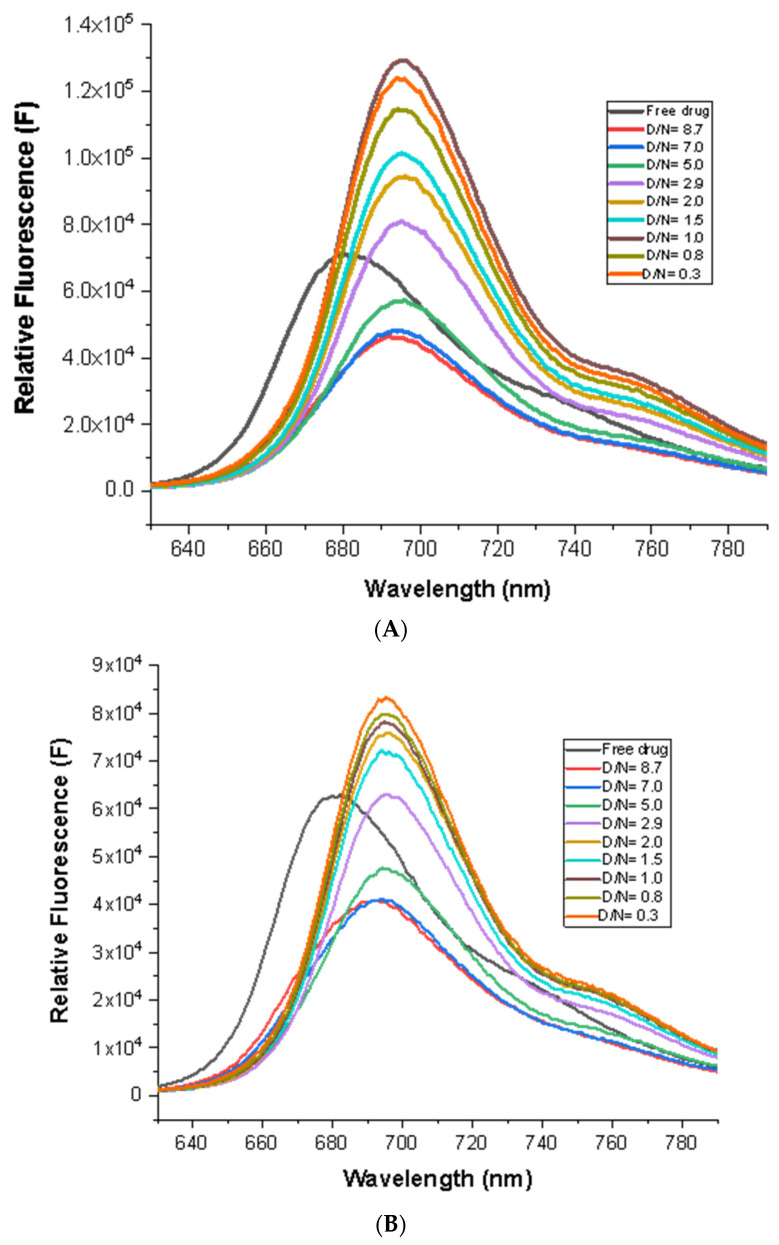
Fluorescence spectra of 7 μM MTX with increasing concentration of G4 DNA at D/N = 0.3–8.7 in 10 mM of KBPES buffer (pH 7.0) containing 100 mM KCl at 25 °C, λ_ex_ = 610 nm: (**A**) *mosR* and (**B**) *ndhA*.

**Figure 5 genes-14-00978-f005:**
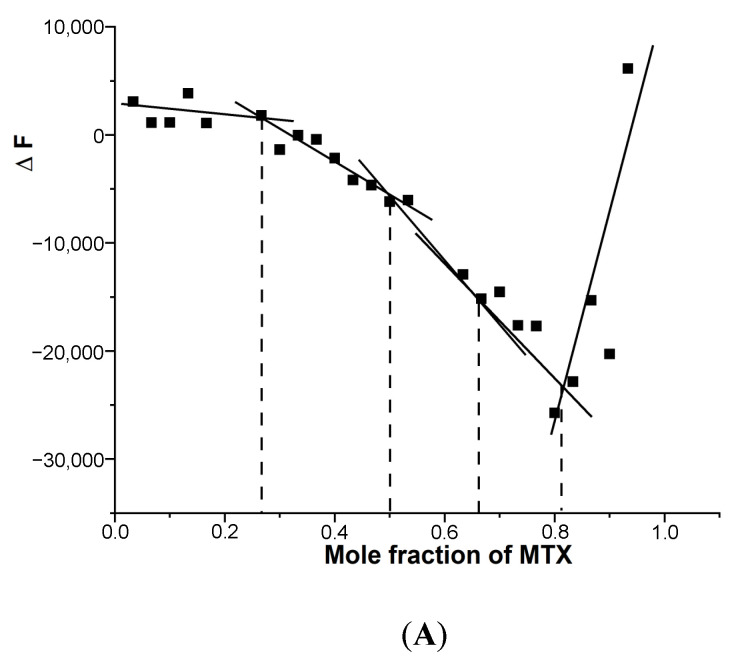
Job plot for the binding of MTX to G4 DNA in 10 mM of KBPES buffer (pH 7.0) containing 100 mM KCl at 25 °C using fluorescence. The total concentration of ligand and DNA is kept constant at 3 μM: (**A**) *mosR* and (**B**) *ndhA*.

**Figure 6 genes-14-00978-f006:**
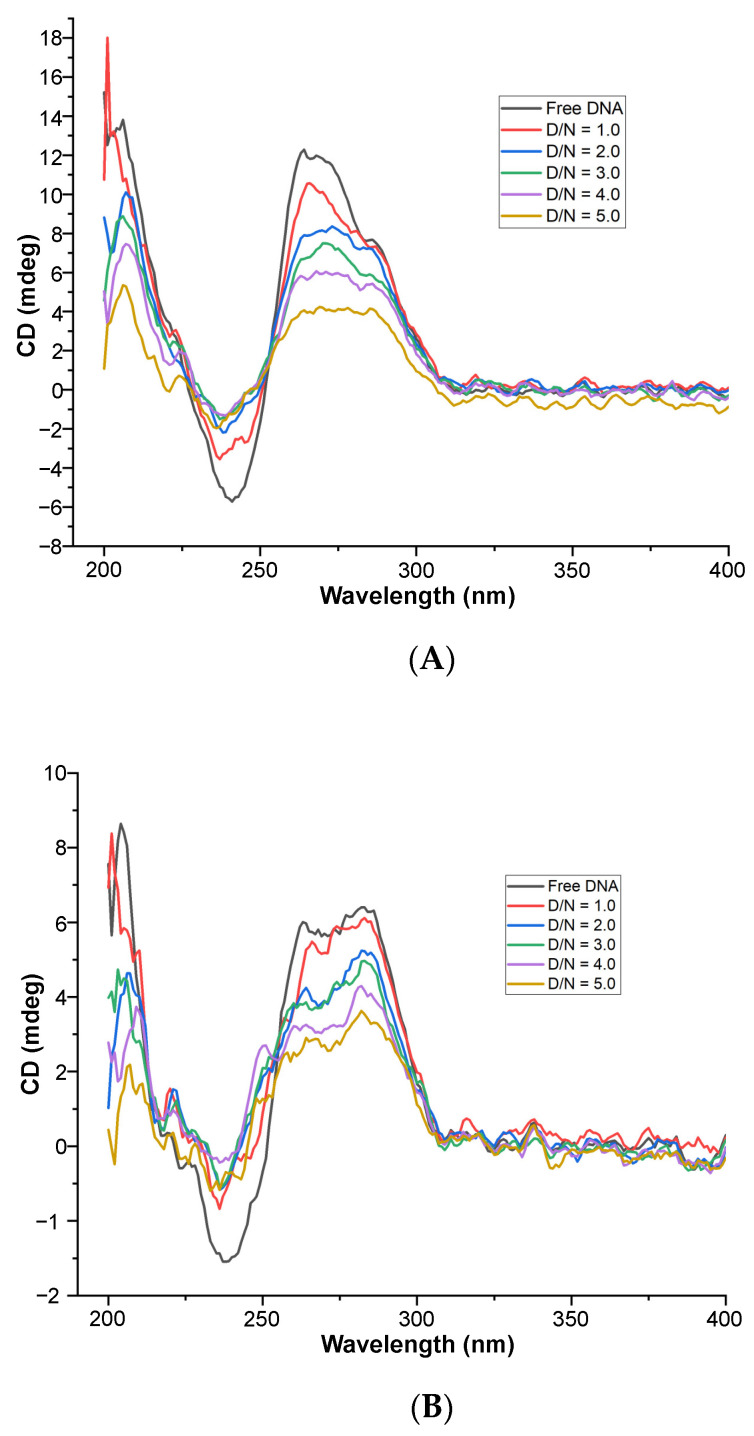
Circular Dichroism spectra of free 20 μM G4 DNA and its complex on the progressive addition of MTX at D/N = 1.0–5.0 in 10 mM of KBPES buffer (pH 7.0) containing 100 mM KCl at 25 °C: (**A**) *mosR* and (**B**) *ndhA*.

**Figure 7 genes-14-00978-f007:**
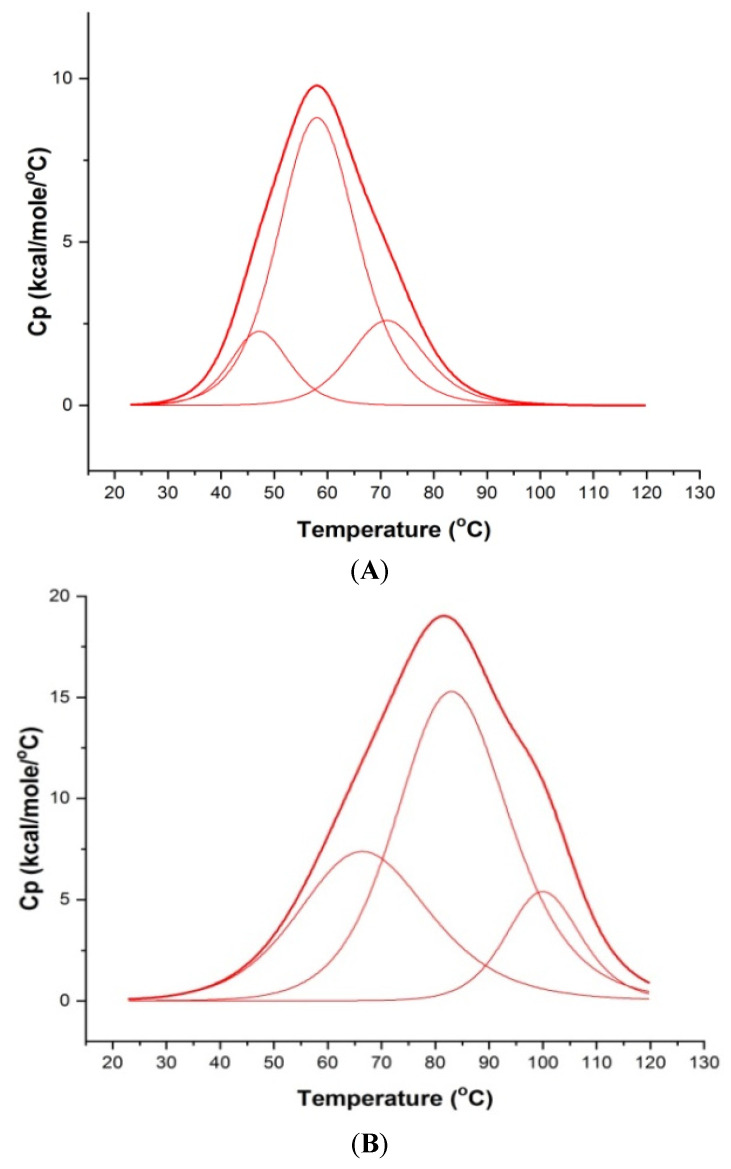
Melting profiles showing excess heat capacity as a function of temperature: (**A**) 100 μM free *mosR*; (**B**) MTX–*mosR* complex at D/N = 5.0; (**C**) 100 μM free *ndhA*; and (**D**) *MTX–ndhA* complex at D/N = 5.0 in 10 mM of KBPES buffer (pH 7.0) containing 100 mM KCl. The observed raw data (thick red lines) in each Figure have been deconvoluted into three “two-state” processes (thin red lines), respectively, using Origin 7.0.

**Figure 8 genes-14-00978-f008:**
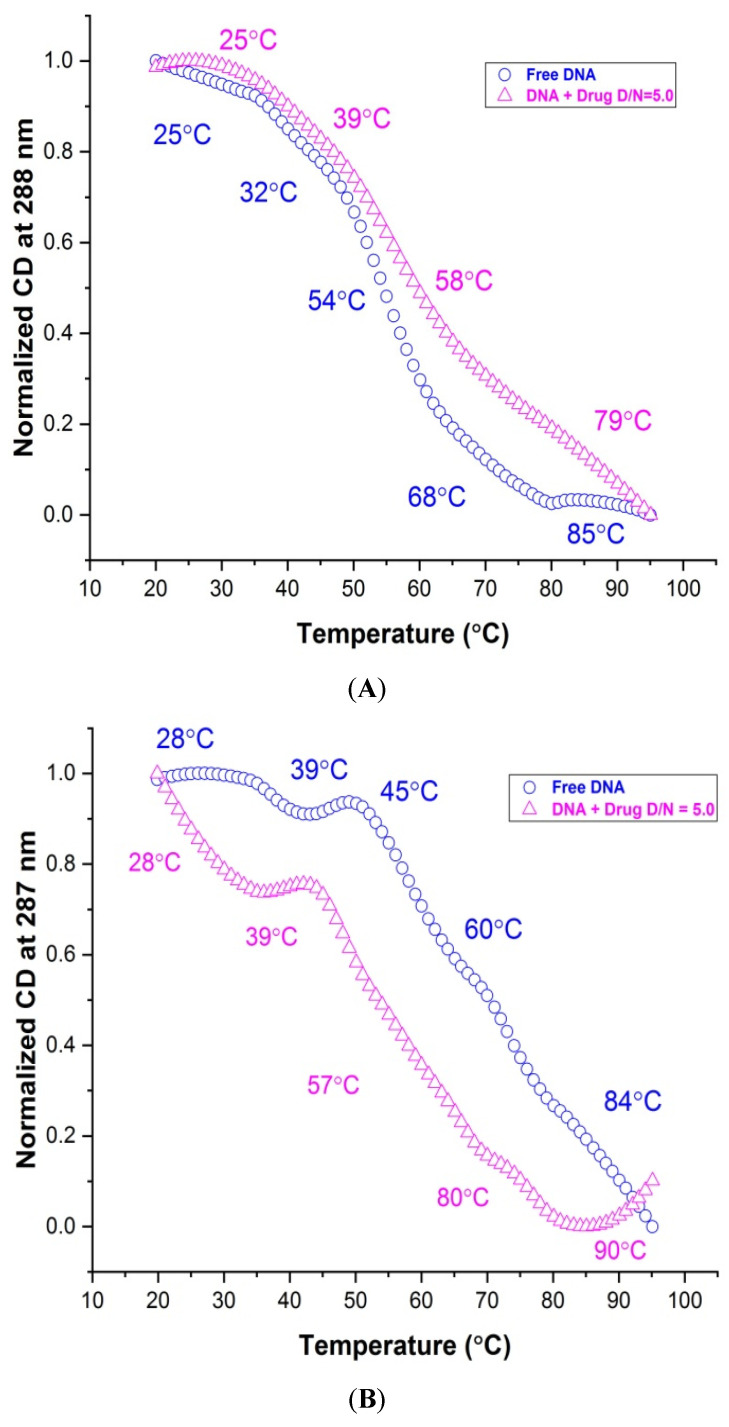
Thermal melting profiles showing normalized CD (in millidegrees) for 20 μM free G4 DNA and its complex with MTX at D/N = 5.0 in 10 mM of KBPES buffer (pH 7.0) containing 100 mM KCl: (**A**) free *mosR* and its complex at 288 nm; (**B**) free *ndhA* and its complex at 287 nm; (**C**) free *mosR* and its complex at 265 nm; and (**D**) free *ndhA* and its complex at 260 nm.

**Figure 9 genes-14-00978-f009:**
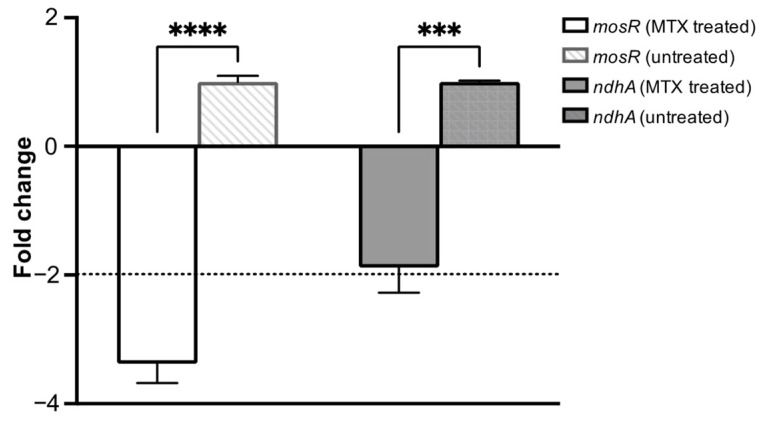
*Mtb* H37Rv was treated with 5XMIC of MTX; RNA was isolated and subjected to qRT-PCR for *mosR* and *ndhA*. qRT-PCR data showing the expression of the genes were normalized with 16S rRNA and compared with the untreated control. The data shown are the result of three independent experiments. Results are expressed as mean ± SD. *** *p* < 0.001 **** *p* < 0.0001.

**Table 1 genes-14-00978-t001:** Melting temperature, *T_m_* (°C), of *mosR* and *ndhA* in free state and that bound to mitoxantrone (MTX) at D/N = 5.0 by fitting the data into three “two-state” transition models using Differential Scanning Calorimetry (DSC). The binding-induced thermal stabilization, Δ*T_m_* (°C), in *mosR*/*ndhA* G4 DNA sequences and χ^2^/Degrees Of Freedom for the goodness of fit are also shown. Symbol * indicates a major component in the DSC thermogram.

Sample	*T* _*m*1_	∆*T*_*m*1_	*T* _*m*2_	∆*T*_*m*2_	*T* _*m*3_	∆*T*_*m*3_	χ^2^/DOF
free *mosR*	47.3 ± 0.6	-	58.3 ± 0.2 *	-	71.4 ± 0.5	-	1.7 × 10^4^
*mosR +* MTX	67.2 ± 1.6	19.9	83.6 ± 0.3 *	25.3	100.2 ± 0.4	28.8	1.9 × 10^5^
free *ndhA*	46.8 ± 0.5	-	64.0 ± 0.4	-	84.4 ± 0.3 *	-	9.9 × 10^4^
*ndhA* + MTX	46.5 ± 0.4	-	81.9 * ± 0.5	17.9	94.5 ± 0.3	10.1	1.4 × 10^5^

## Data Availability

Data supporting results are available in Appendix A.

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
