# Peer review of "MOSR and NDHA Genes Comprising G-Quadruplex as Promising Therapeutic Targets against Mycobacterium tuberculosis: Molecular Recognition by Mitoxantrone Suppresses Replication and Gene Regulation"

_genes, 2023, doi:10.3390/genes14050978_

Round 1

Reviewer 1 Report

In this manuscript Day and coauthors present the data on the effect of Mitoxantrone on Mycobacterium tuberculosis, showing that the compound interacts with two G-quadruplex (G4) structures in the promoters of two bacterial genes.

In general, I find that there are too many assumptions that are not proved by the results.

First, mitoxantrone is an antitumor drug that acts by intercalating into the double-stranded DNA. I do not find any controls showing that mitoxantrone interacts better with the G4 versus the double-stranded DNA.

The authors claim that the G4 motifs contained in the promoter region of mosR and ndhA are involved in the transcriptional regulation of these genes as they are downregulated after treatment with 5X MIC of MTX (as indicated in the materials and methods). This concentration of MTX is very high; consequently, the observed downregulation may be due to a general damage at the genome level.

The authors should check M. tuberculosis viability and repeat the experiment with controls showing that the downregulation observed for mosR and ndhA does not occur in genes whose promoters do not contain G4 motifs. This would be an indication but not a proof that the G4 motifs may play a role in mosR and ndhA transcriptional regulation. To get a stronger evidence, the authors need to disrupt the G4 motifs by site-directed mutagenesis without affecting the promoter consensus sequences, clone the wild-type and mutated promoters in front of a reporter gene and check its expression either by real-time PCR or by measuring the activity of the reporter protein.

Author Response

Reply to reviewer 2 is as per attached file

Reviewer 2 Report

The manuscript deals with interaction of mitoxantrone, an anthraquinone derivative, with G- quadruplex DNA sequences present in mosR and ndhA genes of the Tuberculosis (T.B.) disease causing bacteria, namely, Mycobacterium tuberculosis.

The authors verified formation of stable hybrid G-quadruplex structure in 34/32-mer DNA sequences of these genes by circular dichroism (CD) spectroscopy. SPR sensograms show real-time binding of mitoxantrone to G-quadruplex mosR/ndhA DNA sequences. Interaction leads to hypo-/hyper-chromism, fluorescence quenching and CD changes which have been attributed to partial stacking/groove binding. Binding induced thermal stabilization of these G-quadruplex DNA is substantial, ~18-29 °C. In-vitro experiments show that mitoxantrone binding halts the DNA replication by Taq polymerase enzyme. In vivo studies show significant down-regulation of these genes on treatment of bacteria with mitoxantrone, which will have direct effect on survival of bacteria inside the host. Both DNA replication and gene regulation require single strand stretch of DNA, the stabilization of G-quadruplex DNA formed interferes in their functioning.

The studies demonstrate that treatment of mitoxantrone can halt growth of bacteria and therefore act as anti-tuberculosis drug. Recently Multiple Drug Resistant (MDR) strains of this bacteria have emerged, which cause immense problem in treatment by the existing antibiotics. The genes mosR and ndhA are essential for survival of bacteria since they are involved in oxygen sensing and ATP synthesis processes and are therefore good therapeutic targets for new drugs. In this context present studies are a significant contribution to the continuing search for novel drugs. The results presented,  I believe, may certainly pave the way for development of anthraquinone based drugs for treatment against MDR strains and have therapeutic applications.

Author Response

Response to reviewer 2 comments are as per attached file

Reviewer 3 Report

The authors studies MTX binding to G4 DNA in 2 genes in Mtb and showed its impact on reducing transcription. Overall, the content is interesting, and the results have implication for new drug discovery for Mtb infections.  I think the paper can be published after implementing the following changes. First, the Introduction section should be better organized and needs to introduce prior studies that target G4 DNA, particularly those in Mtb genes, driving to the need / novelty of this study. Second, the authors may consider adding both absorption and fluorescence experiments with a higher D/N (e.g., D/N = 20 or 50) to try to get another data point between “free DNA” and “D/N=7” so that the trend can be seen more clearly. Third, Figure S8 needs a control for the absence of MTX. Finally, there are lots of typos and incorrect grammar throughout the manuscript. The authors should make sure to do a thorough sanity check before re-submitting the paper.

Specific Point-to-Point Comments:

The mosR and ndhA genes involved in pathways of oxidation sensing regulation and ATP generation, respectively, making Mycobacterium tuberculosis (Mtb) bacteria susceptible to oxidative stress inside host macrophage 14 cells.”  --- think here should use “responsible” rather than “susceptible”.

“Additionally, a specific helicase that targets G4s (DinG) and for a G4 aptamer that inhibits a poly-49 phosphate kinase involved in the inorganic polyphosphate intracellular metabolism is 50 present in Mtb” --- This sentence is not relevant.  

“Many dormancy regulon factors controlled most operon which is involved in mycobacterial hypoxic 55 growth.” --- this sentence is coming out of place.

Mycobacterium tuberculosis also shows an anaerobic response after forming the ΔmosR mutant. Under certain stress conditions in the host microenvironment, (e.g., a high Olevel) bacteria go into the log phase and mosR expression gets lower as a resultant relief 58 from the repression of other genes like hypoxia and phagosome expression genes. “---this still doesn’t clearly provide the reasons why mosR is chosen as a target.  

“Ndh and NdhA, both genes have biochemically equal property and have a significant role in ATP synthesis by ATP synthase in the oxidative phosphorylation pathway, which is an essential pathway in Mtb.” – The authors mentioned 2 genes (Ndh and NdhA) here, then why did you target NdhA vs. Ndh? Need a rationale here. Also, need to more explicitly explain HOW these genes play their roles in the ATP synthesis.

“The transcription start site (TSS) of the mosR positioned -38 Nts and ndhA in -47 Nts TSS genes have a high putative G-quadruplex constructing sequence score, making it a propitious therapeutic target for this infection”—Incorrect grammar. Also, need to further explain the “G-quadruplex constructing sequence score” (e.g., how this score was determined, to what extent a high score would imply the existence of a G4 structure)

“both the genes can be assumed as a highly efficient target against the H37Rv deadly strain”---where does the “H37Rv” come from? Why being specific to this strain?

Various ligands reported as G-quadruplex ligands e.g. anthraquinone derivatives, flavonoids, alkaloids, telomastatins, anthracyclines etc. bind either through 72 stacking mode at both DNA ends via favorable…”,--- missing parenthesis before and after “e.g.,…….”. Also, would be better to introduce stackers and groove binders, separately.

The section “3.1. mosR/ndhA DNA sequences adopt stable G4 DNA conformation” should also introduce the feature of CD spectrum for a duplex structure to further demonstrate formation of G4 vs. duplex.

We selected 100 mM K+ as cation concentration in 10 mM KBPES buffer (pH 7.0) throughout our studies since as our focus was on study of interaction with stable solution structures.” ---Wrong grammar. Also, this sentence would be better to be placed to the beginning of the section.

“Analysis of fluorescence data indicated presence of several stoichiometric complexes with n ~1, 2, 4” --- should discuss the 2 complexes separately, and compare the n with the n calculated from the absorption data.

Why does the proximity lead to fluorescence quenching at D/N=5? The MTX should bind “everywhere”, including the preferred binding sites, which should increase the fluorescence as the authors suggested. Did the authors try to say the fluorescence quenching effect from those binding to the non-preferred sites offsets the fluorescence enhancement effect from those binding to the preferred sites at this high D/N ration? The authors need to better describe the explanation / rationale here.

“The observed lifetimes of complexes suggest external binding/partial stacking of MTX to G4 DNA involving contribu tions from side chains” – what do you exactly mean by “external binding / partially stacking”? Does end stacking / groove binding count as a type of “partially stacking” (I assume it does based on what the authors have described in the paper). Then, the authors subsequently made the following statement “It has also been shown that the fluorescence lifetime of porphyrins, TMPyP4 and TrPyP4 (τ = 4.6-5.0 ns in the free form), increases to 6.6-7.5 and 10.6-12.4 ns, respectively and corresponds to end stacking and groove binding modes of interaction with d-(TTGGGGT)4 G4 DNA” to suggest it should not be end stacking / groove binding, as that would increase the fluorescence lifetime. So, the two statements seem contradictory.

“Apparently multiple complexes with stoichiometry of 0.5:1, 1:1, 2:1 and 4:1 exist in solution, which correspond to slope changes at MTX/mole equivalent of G4 DNA fractions of 0.33, 0.50, 0.67, 0.82 and 0.33, 0.5, 0. 82 in mosR and ndhA complexes, respectively.”—The authors should also compare these numbers to the stoichimetry obtained from absorption.

3.6. Stall replication machinery by Taq polymerase enzyme assay” ---This section lacks introduction to the experiment design and how they run the experiment.

direct proof for the involvement of a G4-mediated mechanism in the transcription of the G4 motif-containing genes mosR and ndhA, and they provide a new therapeutic approach for dealing with this lethal human pathogen.”—Need to change the tone, as this study cannot “prove” the statement. Instead, the authors may replace the sentence with something like “the results suggest a therapeutic potential of the drug for Mycobacterium. tuberculosis, potentially through inhibition of transcription by binding to G4 motif.”

Author Response

Response to Reviewer 3 comments are as per attached file

Round 2

Reviewer 1 Report

The authors have made efforts to answer my previous comments and the manuscript has improved.